# Construction and Evaluation of an Optical Medium Voltage Transducer Module Aimed at a 132 kV Optical Voltage Sensor for WAMPAC Systems

**DOI:** 10.3390/s22145307

**Published:** 2022-07-15

**Authors:** Grzegorz Fusiek, Pawel Niewczas

**Affiliations:** Department of Electronic and Electrical Engineering, University of Strathclyde, Glasgow G1 1XW, UK; p.niewczas@strath.ac.uk

**Keywords:** fiber Bragg grating, optical voltage sensor, piezoelectric transducer, capacitive voltage divider, power network instrumentation

## Abstract

This paper reports on the construction and characterization of an optical voltage transducer module for applications in the field of wide-area monitoring, protection, and control (WAMPAC). The optical medium voltage transducer (MVT) module was designed to be combined with a capacitive voltage divider (CVD) to form a voltage sensor intended for 132 kV high voltage (HV) networks. The MVT module comprises a combination of a piezoelectric transducer (PZT) and a fiber Bragg grating (FBG) as a core optical sensing element. Changes in the input voltage across the PZT translate into strain being detected by the FBG. The resultant FBG peak wavelength can be calibrated in terms of the input voltage to obtain a precise voltage measurement. The module was experimentally evaluated in the laboratory, and its performance was assessed based on the requirements specified by the IEC standards for electronic voltage transformers and low power voltage transformers. The results of accuracy tests demonstrate that the MVT module is free from hysteresis, within the experimental error, and is capable of simultaneously meeting the requirements for 0.1 metering and 1P protection classes specified by the IEC 60044-7 and IEC 61869-11 standards.

## 1. Introduction

High energy prices and decreasing cost of distributed energy resources combined with the need for the reduction of greenhouse gas emissions have encouraged utilities to replace coal- and gas-fired power stations with renewable generation [1]. In 2018, about 33% of energy in Europe and the United Kingdom was produced from renewable energy sources (RES) [1]. The European Commission’s long-term plans from 2011 were to increase the energy generation from RES to 66% by 2050, and this objective is still supported today [1,2,3]. The proposed energy transformation pathway is based largely on renewable energy sources and steady improvements in energy efficiency, with wind and solar photovoltaic (PV) technologies envisaged to make the bulk of the European RES installations [1]. This shift toward RES has a high impact on the existing electrical network infrastructure. Increased complexity of the power system presents new challenges to existing wide-area monitoring, protection and control (WAMPAC) schemes [4]. WAMPAC ensures the increased network visibility and fast reaction to the network demands and disturbances, a significant reduction in the number of blackouts, and improvements in the power networks reliability and security [5,6,7]. This is achieved by combining the most recent developments in the fields of conventional measurement, communications, and computing [8].

Electricity distribution networks are essential to wide-area power systems and combine overhead lines, underground cables, and step-down transformers to distribute electrical power to primary and secondary customers [9,10]. Among many problems related to the electrical network functionality and stability are temporary or permanent faults potentially causing damage to electrical equipment and leading to temporary power outages or blackouts with durations sometimes reaching several hours [10,11,12,13]. Therefore, novel sensor systems offering remote and distributed voltage and current measurements while remaining cost-competitive with the current technology are desired for the developing WAMPAC systems.

A suite of optical voltage and current sensors utilizing fiber Bragg gratings (FBGs) and piezoelectric transducers (PZT) was previously proposed by the authors to enable multiple, remote, distributed, passive current and voltage measurements over long distances that can be applied to a wide range of metering and protection applications [14,15,16,17,18]. The proposed variants utilize low-voltage “soft” piezoelectric transducers to either measure the output of a conventional CT or Rogowski coil for current measurement or use the same method for sensing the secondary voltage of a capacitor divider, e.g., to achieve voltage measurement at 132 kV [17]. Alternatively, direct voltage measurement at medium voltage by using larger “hard” piezoelectric (PZT) transducers measuring multi-kV-level voltages were also developed [16,18]. 

Hard piezoelectric materials can be operated at higher voltages and demonstrate better linearity, smaller hysteresis, and better short- and long-term stability than the soft equivalents [10]. Narrower hysteresis in the hard PZTs directly translate into smaller phase displacements between input and output signals of the sensors. This in turn results in no need for hysteresis compensation, saving precious signal processing and computing power required by the interrogator hardware and allowing for interrogating a greater number of sensors. It is also expected that long-term voltage-to-strain response of hard PZTs is superior over the soft PZTs due to the more stable material properties of the hard PZT transducers [19,20,21]. In the previous tests that were carried out by the authors, the effects of aging and de-aging (sensitivity changes) in the soft PZT were worse than those in the hard PZT [10,19,22], which is especially important at rapid signal changes, such as those during faults or lightning impulse events. However, for the application of the hard PZT for direct measurement of high voltage, a scaled-up version of the previous design for MV networks [16,18] does not seem to be feasible. This is because the requirements for the limits of the permissible electric field in the PZT components and the consequent dimensions of the component do not seem to be practical for an HV sensor [10]. To protect the PZT stack against overvoltage conditions, it must be designed to withstand excessive voltage of power frequency and the lightning impulse tests, which requires large piezoelectric transducers. This can be achieved by designing the sensor to operate at voltage levels being only a fraction of the sensor maximum working range, but the size of the component would not be practical in this case. Therefore, a medium-voltage hard piezoelectric transducer (MVT) was proposed by the authors as reported in [10] to work in tandem with a CVD. The proposed MVT is aimed at monitoring 132 kV networks, but it could easily be adapted to other network voltage levels by providing a suitable CVD.

In this paper, we concentrate on the construction and experimental evaluation of the MVT module proposed in [10]. This paper focuses on the assessment of the MVT module’s capability to comply with metering and protection class requirements according to the IEC 60044-7 and IEC 61869-11 standards [23,24]. The analysis presented in this paper considers only the voltage delivered from the CVD output, which is apparent across the MVT terminals. The stability of the CVD division ratio and other effects related to CVD performance are not discussed in this paper [10].

## 2. Materials and Methods

### 2.1. Sensor Voltage Requirements

The proposed HV optical voltage sensor (OVS) is designed to be connected between one phase of a 50 Hz 132 kV system and Earth. Its rated voltage is equal to 76 kV, with a rated voltage factor of 1.2 applicable to measurements between phase and Earth on a continuous basis. The device is expected to withstand the rated power–frequency voltage of 275 kV and the rated lightning–impulse voltage of 650 kV as specified by the relevant IEC standards [23].

### 2.2. Fiber Bragg Gratings Technology

An FBG sensor is a periodic alteration of the refractive index in an optical fiber core, which is formed by exposing a 5–10 mm section of the core to ultraviolet (UV) light of modulated intensity. When broadband light illuminates the FBG, it reflects a range of wavelengths of the incident light with a distinctive peak at so-called Bragg wavelength, *λ_B_*. The Bragg wavelength is determined during production of the sensor and is determined by the grating period, Λ, and the effective refractive index, *n_e_*, of the fiber [25]:(1)$$\lambda_{B}=2n_{e}\Lambda$$

The grating period and the refractive index are functions of temperature and strain affecting the optical fiber. Thus, FBGs can measure both temperature and strain directly or indirectly in other quantities, such as voltage, current, magnetic field. The change in the Bragg wavelength due to change in temperature, Δ*T*, and strain, *ε*, is described by the following equation [25]:(2)$$\frac{\Delta \lambda_{B}}{\lambda_{B}}=k_{T}\Delta T+k_{\varepsilon }\varepsilon$$
where *k_T_* and *k_ε_* are the temperature and strain sensitivities, respectively. By isolating an FBG from the strain, only temperature can be measured. To measure mechanical strain, temperature compensation is required. This is normally achieved by using a dedicated temperature-measuring FBG [10].

The FBGs can be daisy chained to allow for relatively easy interrogation and multiplexing, and their wavelength-encoding nature makes them immune to the problems of intensity fluctuations and attenuation [25].

### 2.3. Medium Voltage Transducer

The MVT design was previously presented in [10]. In the proposed design, the FBG sensor suspended, between two ceramic arms attached to two metallic electrodes bridges, a single cylindrical block of PIC181 material from Physik Instrumente (PI) [10,22,26]. The piezoelectric element and the ceramic arms are attached to the electrodes using thin layers of a conducting epoxy while the fiber is attached to the arms using a UV epoxy. Strain proportional to the input voltage is transferred to the FBG by the piezoelectric component. The MVT construction ensures theoretical two-fold strain amplification [26]. 

The MVT specifications are summarized in Table 1 below.

The sensor is suitable for remote interrogation, and by tracking the instantaneous FBG peak wavelength shifts, the input voltage can be reconstructed. Tracking the average wavelength allows the local sensor temperature to be derived, which can be used for temperature compensation of the sensor voltage readings as discussed in [10].

### 2.4. Sensor Construction

The proposed construction of a medium voltage transducer is shown in Figure 1. The MVT was placed in a hermetic ABS (polyacrylonitrilebutadiene-styrene) enclosure with a polyurethane gasket, and the electrical connections between the electrodes and the relevant connectors were provided using copper wires with a 2 mm diameter. The transducer was then potted with a dielectric gel, having a dielectric strength of 23 kV/mm and dielectric constant of 2.7, to reduce the electric field strength around the sensor. For the duration of the curing process of the gel, the unit was kept in a vacuum to facilitate degassing of the gel. The abovementioned enclosure will be modified for integration with a dedicated CVD. The purpose of the present enclosure is to provide a suitable arrangement for detailed characterization of the prototype device.

Calibration and testing of the transducer module were performed as described in Section 3.

### 2.5. Capacitive Voltage Divider

The anticipated configuration for a high-voltage (HV) capacitive voltage divider is shown in Figure 2 [10]. The capacitive voltage divider is used to reduce the high voltage to the level that can normally be managed by the MVT. To bring the voltage down from 76 to 1 kV, a voltage division ratio (VDR) of 76 was chosen. Assuming an MVT capacitance of 10.4 pF, the practical values of capacitors C_1_ and C_2_ would be 1 and 75.2 nF, respectively.

A bespoke CVD will be housed in a single HV composite insulator and will be provided by an external supplier as in the case of the alternative HV OVS design presented in [17]. Access to the medium voltage output of the divider will be provided inside a suitable enclosure at ground level where the MVT will be placed and connected to the isolated CVD medium voltage terminal [10,17].

The CVD will be required to comply with the requirements for power frequency and the lightning impulse will withstand voltages and partial discharge limits set by the relevant IEC standards. The details of the CVD design are beyond the scope of this paper and are considered as future work.

## 3. Results

### 3.1. Accuracy Requirements

As discussed earlier, the MVT is dedicated to monitor the output of a bespoke HV CVD, and its rated voltage is 1 kV (1.41 kV peak) [10]. For a primary rated voltage of 1 kV, the relevant rated voltage factors are 1.2 and 1.5, which are applicable to measurements between phase and Earth on a continuous and 30 s duration basis, respectively [23]. The device must also withstand the rated power frequency for 60 s and 30 positive and negative impulses of the rated lightning impulse voltage levels, as set by the relevant standards.

Since the sensor is aimed at being compliant with protection and metering classes, its voltage measurement errors at the rated frequency must be below the limits specified in Table 2 and Table 3 [23,24].

For metering class, the voltage and phase errors at the rated frequency should not exceed the values specified in Table 3 at any voltage between 80% and 120% of the rated voltage [18,23,24].

In addition to the above requirements, the accuracy needs to be verified at the rated voltage and at frequencies equal to 96% and 102% of the rated frequency, the requirement for protection class devices, and at frequencies equal to 99% and 101% of the rated frequency, the requirement for metering class devices [18,23,24].

### 3.2. Experimental Setup

A schematic diagram of the experimental setup is shown in Figure 3. Calibration and testing of the MVT module was performed by applying a comparison method specified by IEC 60060-2 [27]. The device under test (DUT) was connected in parallel with a voltage reference system and was calibrated by comparing both measurement system outputs.

The prototype MVT was calibrated and tested in the laboratory conditions at room temperature (20 ± 1 °C). The test voltage was provided from a step-up transformer (Majestic Transformers) with a voltage ratio of 400/20,000 V/V. The transformer was powered from a programmable AC source, Chroma 61512, capable of delivering 18 kVA at 300 V. A Fluke 80K-6 voltage probe (VP), having a 1000:1 V/V voltage division ratio (VDR) and an AC accuracy of ±1%, was used as a voltage reference. The unit is suitable for measuring DC and AC voltages up to 6 kV. The VP output was monitored by a dedicated PCIe 6363 (National Instruments) data acquisition card (DAQ) installed in a personal computer (PC).

During calibration and accuracy testing, the MVT sensor was illuminated by a broadband light source, and the reflected signals were analyzed using an FBG interrogator, I-MON 256 USB (Ibsen Photonics), connected to a PC. The optical system clock in I-MON (4 kHz) was used to synchronize the interrogator readings with the DAQ readings. The clock signal from I-MON was provided to the DAQ via a dedicated synchronization link, and the DAQ sampling rate was set to 4 kS/s. The measurements retrieved from both optical and electrical measurement systems were used to calculate rms values and measurement errors. The voltage amplitude and phase errors were calculated according to IEC 61869-11 and IEC 60060-2 [24,27].

A photograph of the MVT module connected to a 400/20,000 V/V transformer and the voltage probe during tests is shown in Figure 4.

The MVT module was energized from a single phase of a 20 kV three-phase transformer. 

### 3.3. Sensor Characterization and Accuracy Testing

To calibrate the MVT, a 50 Hz sinusoidal voltage was applied to the sensor in the range from 2% to 120% of the nominal voltage (1 kV) at 2%, 5% and in 10% steps between 10% and 120% of the nominal. The readings of the optical signals from the sensor and the readings from the voltage probe were saved together at each voltage level. At each voltage level, 400 samples, equivalent to five periods of 50 Hz signals, were captured. The rms values were then calculated from five periods at each voltage level.

The instantaneous reference voltage and wavelength signals at the sensor-rated voltage of 1 kV are shown in Figure 5, and the sensor hysteresis for voltages between 2% and 120% of the rated voltage are shown in Figure 6. As can be seen, there is no hysteresis, and the phase displacement between the wavelength and the reference voltage is nearly zero (within the experimental error).

The sensor calibration curve, shown in Figure 7, was created using a second-order polynomial fitted into the rms data obtained from the first characterization run for the reference voltage and the wavelength between 2% and 120% of the nominal voltage. The resultant parabolic equation was then implemented in the interrogator software and was used to convert the FBG peak wavelength shifts into the measured voltage levels.

To evaluate the capability of the sensor to meet protection and metering accuracy requirements, 50 Hz voltage waveforms with amplitudes of 2%, 20% and between 80% and 120% of the device-rated voltage were applied as part of the test procedure. The measurements were repeated three times. 

The amplitude (voltage) error in steady-state condition is defined according to the following equation:(3)$$\varepsilon_{u}\left(\%\right)=\frac{U_{p}-U_{rec}}{U_{p}}\times 100$$
where *U_p_* is the rms value of the primary voltage, and *U_rec_* is the rms value of the reconstructed voltage.

The phase error is calculated as the difference in phase between the secondary output (reconstructed voltage) phasor and the primary voltage phasor:(4)$$\varepsilon_{u}\left(\%\right)=\frac{U_{p}-U_{rec}}{U_{p}}\times 100$$
where *φ_p_* and *φ_s_* are respectively the primary and secondary phase displacements.

The voltage amplitude and phase errors for the combined three consecutive test runs are shown in Figure 8 and Figure 9, respectively.

The succeeding experiment relied on testing the sensors performance at the rated voltage and at frequencies equal to 96% and 102% of the rated frequency, the requirement for protection class devices, and at frequencies equal to 99% and 101% of the rated frequency, the requirement for metering class devices. For 50 Hz rated signals, this is equivalent to 48 and 51 Hz, and 49.5 and 50.5 Hz, respectively. The voltage amplitude and phase error results of three consecutive runs are presented in Figure 10 and Figure 11.

## 4. Discussion

As can be seen from the results presented in Section 3, the MVT module has minimal hysteresis and nearly zero phase displacement between the input and output signals. The minimal hysteresis of the MVT is highly beneficial, as running intensive signal processing tasks on the interrogator to compensate for hysteresis [28] are not needed. The previous embodiment of the HV OVS based on soft piezoelectric materials required hysteresis compensation to minimize the phase and amplitude errors to meet the measurement accuracy requirements. In the case of the MVT module, by using only amplitude scaling, a large amount of signal processing and computing power is saved in the interrogator hardware, which is especially important when several dozens of sensors are to be interrogated simultaneously.

As indicated in Table 1 and Table 2, for the IEC 0.1 metering class, the voltage errors at the rated frequency should not exceed ±0.1% at any voltage between 80% and 120% of the rated voltage (0.8 and 1.2 kV, respectively) while the phase errors should be below ±0.08° (±5 min). As can be seen from the results presented in Section 3, the MVT voltage and phase errors are within the limits set by the IEC 60044-7 and IEC 61869-11 standards. The voltage and phase errors remained within the limits for the rated voltage and frequencies between 99% and 101% of the rated frequency (50 Hz) as set by IEC 60044-7.

The voltage errors at 2% of the nominal voltage (0.02 kV) were below the limit of ±4%. At 20% of the nominal voltage (0.2 kV), the errors were below the ±2% limit, and between 80% and 120% of the rated voltage (at 1.2 voltage factor for this sensor), the errors were below ±1%. The phase errors at 2% of the nominal voltage (0.02 kV) were below the limit of ±2.66° (±160 min). At 20% of the nominal voltage (0.2 kV), the errors were below ±1.33° (±80 min) limit, and between 80% and 120% of the rated voltage (at 1.2 voltage factor for this sensor), the errors were below ±0.66° (±40 min). Clearly, the MVT performance is better than the requirements set by the IEC 61869-11 standard for the 1P class multipurpose devices, which also satisfies the requirements for the 3P class protection devices. The voltage and phase errors remained within the limits for the rated voltage and frequencies between 98% and 102% of the rated frequency (50 Hz), as required by IEC 60044-7.

Based on the results demonstrated above, it can be concluded that the same device has the potential to meet the 1P protection class and 0.1 metering class.

The measurement accuracy requirements for the proposed device were based on IEC 60044-7 for Electronic Voltage Transformers and IEC 61869-11 for Low Power Voltage Transformers as the most appropriate standards. Since the new IEC 61869-7 standard is not available at the time of writing of this paper, and the document forecast publication date is currently set to December 2022 [22], IEC 60044-7 is still in force.

The MVT calibration and accuracy testing were carried out within a few hours on a single day. Although the divider was sufficiently stable over that period, further work is required to establish its performance in terms of day-to-day or long-term stability.

It is also envisaged that the measurement errors could be further reduced when a more accurate reference voltage probe is utilized for characterization of the sensors, together with an interrogator system offering lower noise level and better wavelength resolution. This could be achievable with the interferometric interrogation method such as that previously proposed by the authors in [29].

## 5. Conclusions

In this paper, the construction and initial characterization of an optical voltage sensor for application in the field of wide-area monitoring, protection, and control (WAMPAC) have been presented. The optical medium voltage transducer (MVT) module under investigation was designed to be combined with a capacitive voltage divider (CVD) intended for 132 kV high voltage networks. The MVT module was experimentally evaluated in the laboratory conditions, and its performance was assessed based on the operational requirements specified by the IEC standards for electronic voltage transformers and low power voltage transformers. The results of the accuracy tests demonstrate that the MVT module is capable of meeting the requirements for the 0.1 metering and 1P protection and multipurpose devices specified by the IEC 60044-7 and IEC 61869-11 standards.

Future work will focus on performing additional MVT module tests, including temperature, routine and special tests (such as lightning impulse tests) according to the relevant standards.

## Figures and Tables

**Figure 1 sensors-22-05307-f001:**
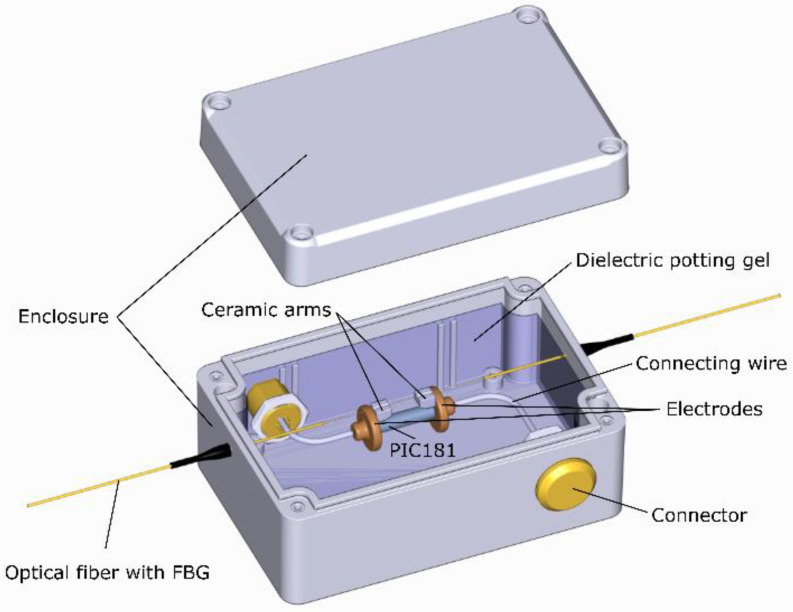
A medium voltage transducer packaged into an ABS enclosure and potted with a dielectric gel.

**Figure 2 sensors-22-05307-f002:**
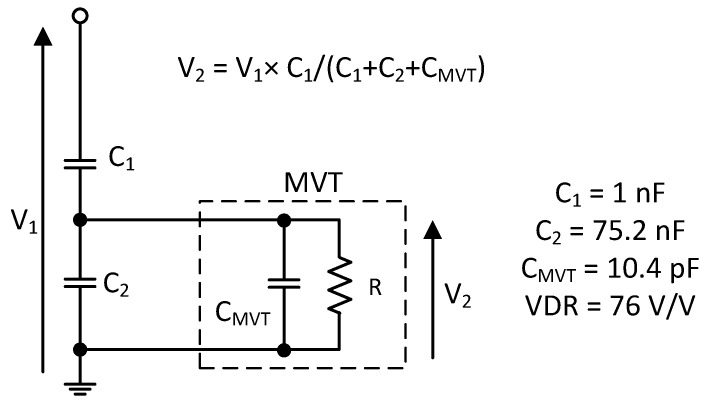
A diagram of the HV-to-MV voltage divider.

**Figure 3 sensors-22-05307-f003:**
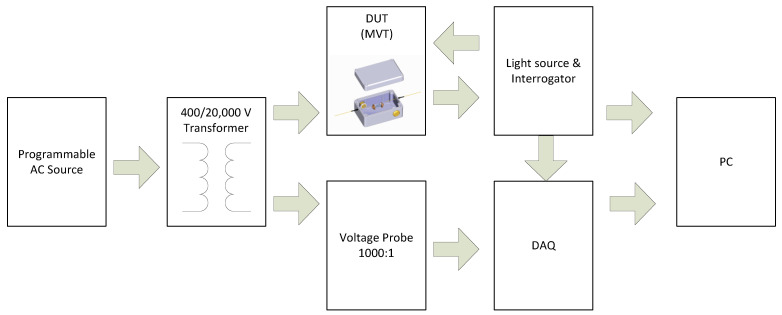
Experimental setup for MVT calibration and testing.

**Figure 4 sensors-22-05307-f004:**
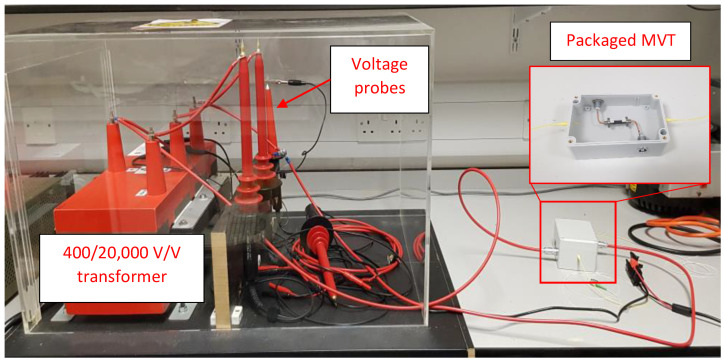
The MVT module connected to a single phase of a 20 kV three-phase transformer.

**Figure 5 sensors-22-05307-f005:**
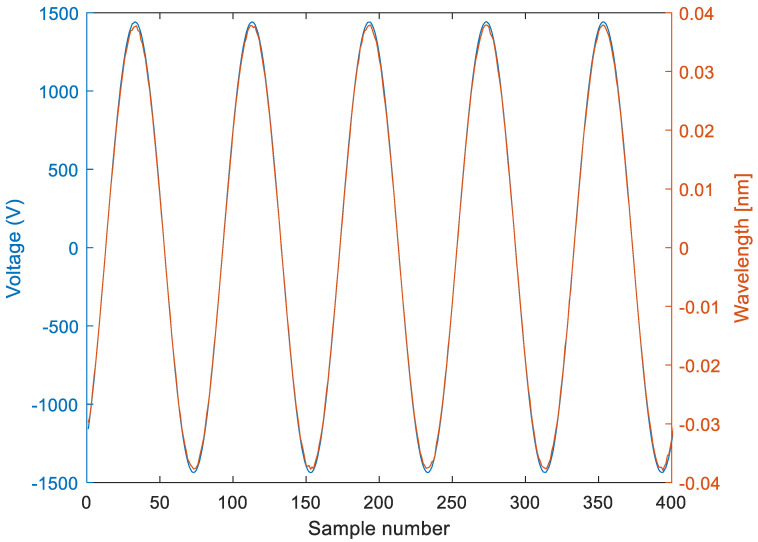
Instantaneous wavelength and reference voltage of the tested MVT module.

**Figure 6 sensors-22-05307-f006:**
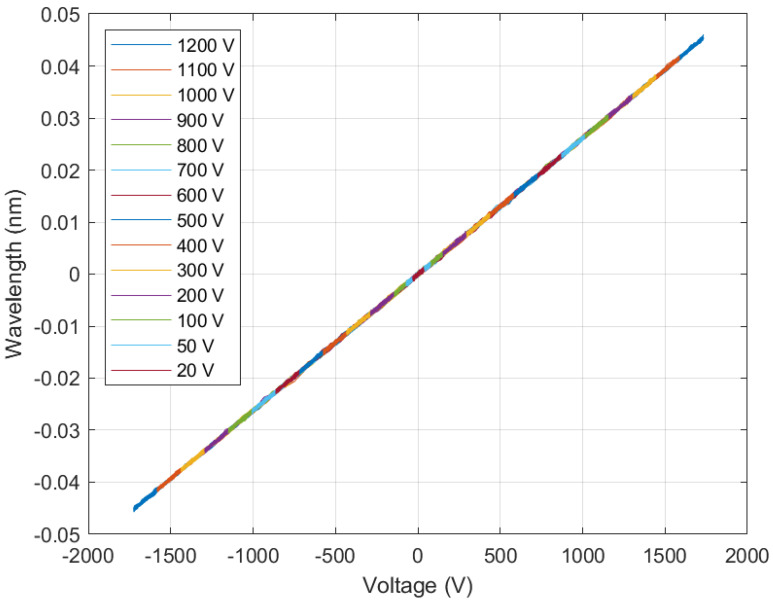
MVT hysteresis. There is no hysteresis between the wavelength and the reference voltage (within the experimental error).

**Figure 7 sensors-22-05307-f007:**
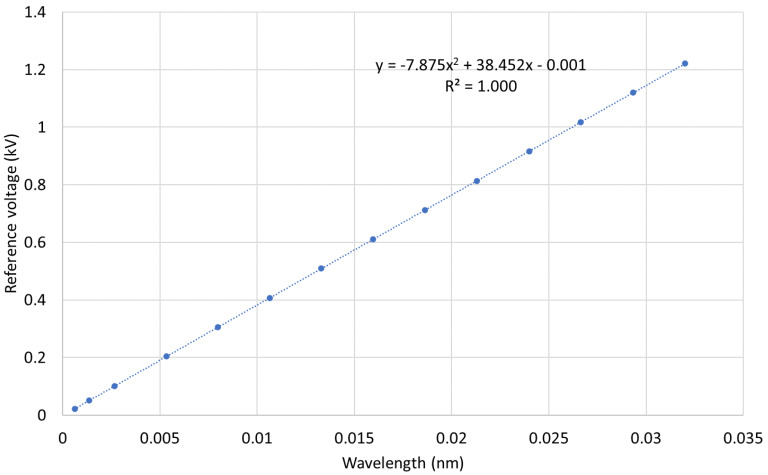
MVT calibration curve.

**Figure 8 sensors-22-05307-f008:**
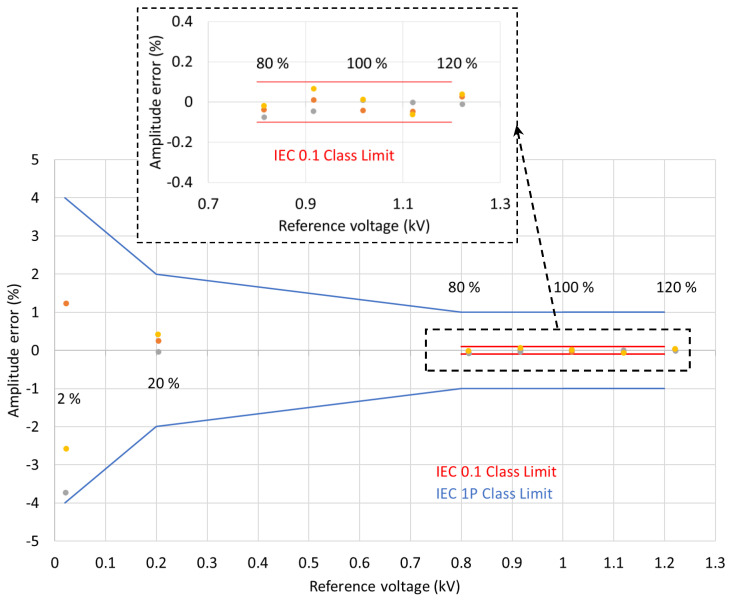
MVT voltage amplitude errors.

**Figure 9 sensors-22-05307-f009:**
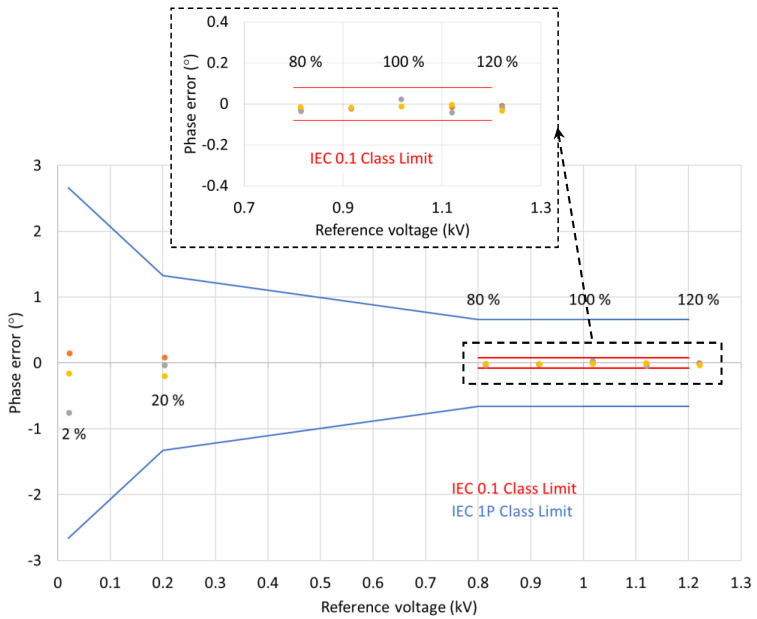
MVT phase errors.

**Figure 10 sensors-22-05307-f010:**
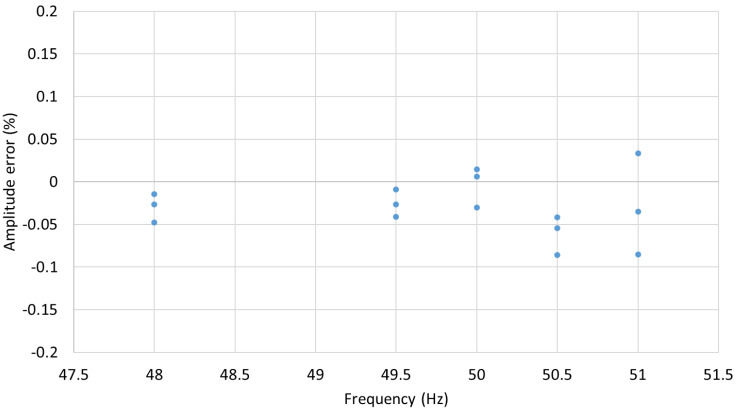
MVT voltage amplitude errors at nominal voltage and frequencies 48, 49.5, 50.5, 51 Hz.

**Figure 11 sensors-22-05307-f011:**
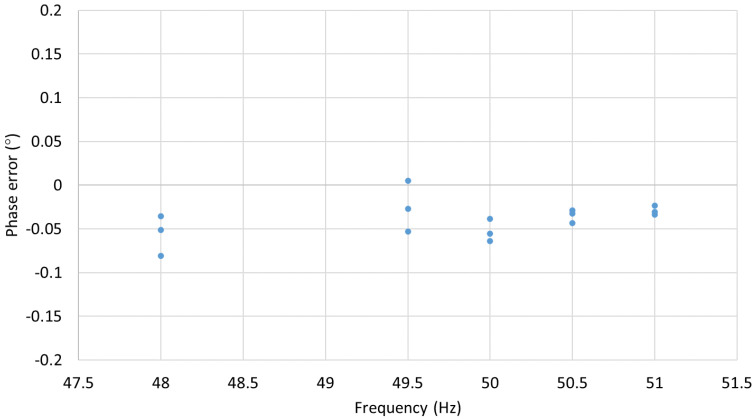
MVT phase errors at nominal voltage and frequencies 48, 49.5, 50.5, 51 Hz.

**Table 1 sensors-22-05307-t001:** Specifications of the MVT module [10,16,18,22,26].

Property	Value
PZT material	PIC181
Length (mm)	20
Diameter (mm)	5
Piezoelectric charge constant d_33_ (pm/V)	265
Resistance R (MΩ)	200
Capacitance C_MVT_ (pF)	10.4
Series resonant frequency (kHz)	80
Full displacement time (µs)	4.2
Maximum permissible electric field strength of the material (kV/mm)	2.5
Maximum compressive stress (MPa)	100
Maximum tensile stress (MPa)	10
Nominal (rated) voltage (kV)	1

**Table 2 sensors-22-05307-t002:** Protection class accuracy requirements as per IEC 61869-11 [24].

Accuracy Class	Voltage (Ratio) Error ε_u_					Phase Error φ_e_									
	±%					±Minutes					±Centiradians				
	at Voltage (% of Rated)					at Voltage (% of Rated)					at Voltage (% of Rated)				
	2	20	80	100	X ^1)^	2	20	80	100	X ^1)^	2	20	80	100	X ^1)^
0.1P	0.5	0.2	0.1	0.1	0.1	20	10	5	5	5	0.6	0.3	0.15	0.15	0.15
0.2P	1	0.4	0.2	0.2	0.2	40	20	10	10	10	1.2	0.6	0.3	0.3	0.3
0.5P	2	1	0.5	0.5	0.5	80	40	20	20	20	2.4	1.2	0.6	0.6	0.6
1P	4	2	1	1	1	160	80	40	40	40	4.8	2.4	1.2	1.2	1.2
3P	6	3	3	3	3	240	120	120	120	120	7	3.5	3.5	3.5	3.5
6P	12	6	6	6	6	480	240	240	240	240	14	7	7	7	7

^1)^ X is the rated voltage factor multiplied by 100.

**Table 3 sensors-22-05307-t003:** Metering class accuracy requirements as per IEC 60044-7 and IEC 61869-11 [18,23,24].

Accuracy Class	Percentage Voltage (Ratio) Error ±ε_u_ %	Phase Error ±φ_e_	
		Minutes	Centiradians
0.1	0.1	5	0.15
0.2	0.2	10	0.3
0.5	0.5	20	0.6
1.0	1.0	40	1.2
3.0	3.0	Not specified	

## Data Availability

All data underpinning this publication are openly available from the University of Strathclyde KnowledgeBase at https://doi.org/10.15129/5f1cdedf-1bb3-49c3-9085-b41db49b85af (accessed on 7 July 2022).

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
