# Peer review of "Construction and Evaluation of an Optical Medium Voltage Transducer Module Aimed at a 132 kV Optical Voltage Sensor for WAMPAC Systems"

_sensors, 2022, doi:10.3390/s22145307_

Round 1
Reviewer 1 Report
Manuscript No: sensors-1766275
Title: Construction and evaluation of an optical medium voltage 2 transducer module aimed at a 132-kV optical voltage sensor for 3 WAMPAC systems
Authors: Grzegorz Fusiek, and Pawel Niewczas
A. Overview
1. In this manuscript the authors report experimental work on construction and characterization of an optical voltage transducer module for applications in the field of wide-area monitoring
2. The contents are expressed clearly; the manuscript is well organized, and it is written in reasonable English.
3. The authors have acknowledged recent related research.
4. As long as my knowledge, the work presented is original.
B. Detailed analysis.
- Abstract: It is too short. State the novelty of your work. Please organize the ideas in each paragraph.
- 1. Introduction – it provides a state of the art on the subject and there are recent references.
- 2. Materials and Methods
- Before describing the methodology, model and supplier of the equipment’s used should be described, as well for the discrete components.
- Figure captions must be self-explanatories.
- Figure 6 should state that there is no MVT hysteresis.
- Figure 7. MVT calibration curve – there are to many digits in the equation and R2.
C. Overall assessment
In my opinion the work can be published after minor corrections in the manuscript.
D. Review Criteria
1. Scope of Journal
Rating: Medium
2. Novelty and Impact
Rating: Medium
3. Technical Content
Rating: Medium
4. Presentation Quality
Rating: Medium
Reviewer 2 Report
This manuscript addresses the construction and evaluation of an optical medium voltage transducer module aimed at a 132-kV optical voltage sensor for WAMPAC systems. This is a very interesting paper, particularly due to the increasing interest that energy resource distribution and monitoring has nowadays, fed by inflation and politics worldwide. In this work, the authors design an optical MVT module combined with a CVD to develop a voltage sensor aimed at 132-kV high voltage operation. The text provides a proper Introduction, followed by the sensor design procedure. After that, the Results and Characterization deeply covers the sensor features and their compliance with IEC standards. The manuscript is well-written and provides a lot of details on every step of the process. Regarding the presentation, it includes clear pictures, schemas and figures, illustrating the key concepts discussed. The Results are outstanding as well and a Discussion section is dedicated to highlighting the achievements, including the simultaneous compliance with IEC-60044-7 and IEC 61869-11 standards. This is definitely a very good work and an excellent match for MDPI Sensors. I just have a couple of minor suggestions which I recommend to be considered before the acceptance of this work for publication:
- As this work might be read by scientist with an Electrical Engineering background but with limited knowledge of optics, I recommend to add a couple of sentences introducing fiber Bragg gratings. At least mention what they are, their main advantages and that they are typically sensitive to strain and temperature effects. I recommend to add a couple of references pointing to some recent developments of FBG applications, such as Measurement 166, 108229 (2020)(https://www.sciencedirect.com/science/article/abs/pii/S0263224120307685) & Sensors and Actuators A: Physical 332, 113061 (2021)(https://www.sciencedirect.com/science/article/abs/pii/S0924424721005264).
- In Section 3.2, the optical interrogation setup is not clear to me. The authors say that “[…]the MVT sensor was illuminated by a broadband light source and the reflected signals were analyzed using an FBG interrogator, I-MON 256 USB[…]”. From this sentence, I assume that the light source illuminates the sensor and then the reflected signal passes to the interrogator through an optical circulator, therefore using just a single port. However, I see that the sensor illustrated in figure 1 has an optical fiber input and output, being 2 ports. So I was wondering if the measured signal is not the reflected signal but the transmitted one. When I look into the I-MON 256 USB product site, though, it seems that this interrogator operates in reflection mode. So I find this part a bit confusing. Could the authors clarify it?
